# Operando spectroscopy study of the carbon dioxide electro-reduction by iron species on nitrogen-doped carbon

Chiara Genovese[1], Manfred E. Schuster[2], Emma K. Gibson[3,4], Diego Gianolio [5], Victor Posligua[6], Ricardo Grau-Crespo [6], Giannantonio Cibin [5], Peter P. Wells[5,7], Debi Garai[5], Vladyslav Solokha[5], Sandra Krick Calderon[8], Juan J. Velasco-Velez[9], Claudio Ampelli[1], Siglinda Perathoner[1], Georg Held[5,6], Gabriele Centi [10] & Rosa Arrigo [5,6]

The carbon–carbon coupling via electrochemical reduction of carbon dioxide represents the biggest challenge for using this route as platform for chemicals synthesis. Here we show that nanostructured iron (III) oxyhydroxide on nitrogen-doped carbon enables high Faraday efficiency (97.4%) and selectivity to acetic acid (61%) at very-low potential (−0.5 V vs silver/silver chloride). Using a combination of electron microscopy, operando X-ray spectroscopy techniques and density functional theory simulations, we correlate the activity to acetic acid at this potential to the formation of nitrogen-coordinated iron (II) sites as single atoms or polyatomic species at the interface between iron oxyhydroxide and the nitrogen-doped carbon. The evolution of hydrogen is correlated to the formation of metallic iron and observed as dominant reaction path over iron oxyhydroxide on oxygen-doped carbon in the overall range of negative potential investigated, whereas over iron oxyhydroxide on nitrogen-doped carbon it becomes important only at more negative potentials.

[1] Department of Chemical, Biological Pharmaceutical and Environmental Sciences, University of Messina, ERIC aisbl and CASPE/INSTM, V.le F. Stagno d'Alcontres, 31, 98166 Messina, Italy. [2] Johnson Matthey Technology Centre, Reading, RG4 9NH, UK. [3] UK Catalysis Hub, Research Complex at Harwell (RCaH), Harwell, Oxfordshire OX11 0FA, UK. [4] Department of Chemistry, UCL, 20 Gordon Street, London, WC1 0AJ, UK. [5] Diamond Light Source Ltd., Harwell Science and Innovation Campus, Didcot, OX11 0DE, UK. [6] Department of Chemistry, University of Reading, Whiteknights, Reading, RG6 6AD, UK. [7] School of Chemistry, University of Southampton, Southampton, SO17 1BJ, UK. [8] FAU Erlangen-Nürnberg, Egerland Str. 3, 91058 Erlangen, Germany. [9] Max-Planck Institut für Chemische Energiekonversion, Stiftstr. 34 – 36, 45470 Mülheim an der Ruhr, Germany. [10] Department of Mathematical, Computer, Physical and Earth Sciences - University of Messina, ERIC aisbl and CASPE/INSTM, V.le F. Stagno d'Alcontres 31, 98166 Messina, Italy. Correspondence and requests for materials should be addressed to R.A. (email: rosa.arrigo@diamond.ac.uk)

One of the current grand challenges in chemical science is moving towards a solar-driven chemistry, through the conversion of recycled $CO_2$ to chemicals using renewable energy[1,2]. As a consequence, the electrochemical $CO_2$ reduction reaction ($CO_2RR$) over different electrodes[3] is increasingly investigated with the biggest challenge being the formation of products >C1. Copper has shown an incomparable efficiency to form hydrocarbons[4,5], but despite this, a poor stability, selectivity, and high overpotentials are limiting factors. The opportunity to facilitate one selective path amongst the others relies on the possibility to kinetically control the energetics of adsorbed reaction intermediates[6], on a specific surface structure. This demands new ideas in catalyst design attainable through a molecular level understanding of the reaction mechanism[5,6].

With respect to the selectivity issue, molecular catalysis can be product specific and highly efficient, however redox processes are limited to the transfer of only a few electrons, thereby leading to products of lower technological interest such as CO.

Recently, it was shown that the immobilization of molecular species such as Co porphyrins on graphite and graphene opens up opportunities for multistep reduction products[6].

We have also demonstrated the synthesis of acetic acid via $CO_2RR$ over Cu on carbon nanotubes (Cu/CNTs) electrodes[7]. With respect to the multistep synthesis of acetic acid from fossil fuels, the direct $CO_2RR$ allows lowering of the carbon footprint by a factor of 5–6, due to the combination of process intensification, use of $CO_2$ as raw material and use of renewable energy[7].

Nitrogen species in carbon have been also reported to convert $CO_2$ to C1 products, such as $CO$[8,9]. Compared to metal-free nanocarbons, C supported metal nanoparticles allow improving performances and lowering $CO_2RR$ overpotentials[10], but the competing hydrogen evolution reaction (HER) reaction is also favored. Liu et al.[11] reported high $CO_2RR$ efficiency to acetate and formate over Si/N-doped nanodiamond with high Faraday efficiency of 91.2–91.8% at −0.8 to −1.0 V vs RHE, where the high overpotential for the HER was the favorable factor.

In this work, we explore the $CO_2RR$ activity of Fe oxyhydroxide nanostructures supported on O- and N-doped graphitic supports in a $CO_2$-saturated 0.05 M $KHCO_3$ solution. We report the outstanding performance of ferrihydrite-like ($Fh$-FeOOH) clusters on N-doped carbon (N-C) with a total $CO_2RR$ Faraday efficiency above 97 % and high selectivity to acetic acid at very-low potential (−0.5 V vs Ag/AgCl). We apply operando hard X-ray absorption fine structure (XAFS) spectroscopy to obtain insights into: the nature of the sites responsible for $CO_2RR$ at low potentials, particularly those enabling C–C coupling (to form acetic acid); and dynamic structural changes upon potential changes. This study reveals the reversible redox chemistry of $Fh$-FeOOH nanostructures on N-C in low concentration bicarbonate solution, characterized by the formation of Fe(II) species at potentials relevant for $CO_2RR$, whereas at more negative potentials those species turn into $Fe^0$. In contrast, there is no significant formation of Fe(II) species in the $Fh$-FeOOH supported on O-containing carbon in this voltage range, and the only structural modification observed is the reduction of some of the $Fh$-FeOOH clusters to $Fe^0$. The $H_2$ evolution is indeed correlated to the transformation of Fe(III) into $Fe^0$. By a combination of ambient pressure soft X-ray photoelectron spectroscopy (XPS) and density functional theory (DFT) simulations we prove that a chemical interaction occurs between Fe sites of ferrihydrite and the pyridine N species on the carbon surface. As a consequence of the favorable Fe–N interaction, Fe species, initially present as single atoms or clusters decorating the N-functionalized edges of the graphitic planes, are stabilized as Fe(II) species at a potential consistent with the carbonation of ferrihydrite and the formation of a Fe(II)Fe(III) mixed compound[12]. This potential range coincides with the highest Faraday efficiency to $CO_2RR$ products. We conclude that the few relevant species for C–C coupling are an ensemble of chemically interacting (bi)carbonate-bearing Fe (II) species and N atoms, the latter one also capable of chemisorbing $CO_2$-related species[13]. This study deepens our understanding of the reactivity of this class of electrocatalysts in $CO_2RR$ and their structural transformation into HER selective materials and provides guidance for the synthesis of improved electrocatalysts for the $CO_2RR$.

## Results

**Structure of Fe/N-O and Fe/O-C.** In this work, the catalysts were synthesized by impregnation and subsequent thermal annealing of the Fe nitrate precursor on pieces of N and O functionalized C paper. The oxygen functionalized support (O-C) contains mainly carboxylic functional groups[14], whereas the nitrogen functionalized support (N-C) contains mainly pyridine-like N species (Supplementary Fig. 3)[14]. If not otherwise stated, the nominal Fe loading was 1 wt. %, which was quantitatively loaded onto the supports.

The structural characterization of the as synthetized samples was performed by means of XAFS spectroscopy, XPS, scanning electron microscopy (SEM) and transmission electron microscopy (TEM). Fig. 1 reports the ex situ X-ray absorption near edge structure (XANES) spectra at the Fe K edge of all the samples. The positions of the absorption pre-edge (1s → 3d transition) and edge (1s → 4p transition) resonances are sensitive to the Fe oxidation state, whereas the intensity of the pre-edge peak depends on site symmetry, where the lower the intensity the higher the symmetry of the Fe sites. The pre-edge appears at ca. 7115 eV for Fe(III) species as in $Fe_2O_3$, and ca. 7112.5 eV for Fe (II) species as in Fe(II)acetate[15]. Moreover, the pre-edge will be more intense for tetrahedral and distorted octahedral geometries than for octahedral systems.

Consistently, the absorption pre-edge and edge found for both samples at approx. 7114.5 eV and 7125 eV, respectively, hints at Fe(III) species[16]; however the difference with respect to the Fe (III) coordination environment in $Fe_2O_3$ hematite and $Fe_3O_4$ magnetite is significant, as shown in Fig. 1.

The position of the white line at 7132.2 eV and the additional peak at 7147.6 eV have been observed for ferrihydrite[16] ($Fh$-FeOOH), which is an $hcp$ form of Fe oxyhydroxide, where Fe(III) cations are coordinated with O atoms and terminal OH species in

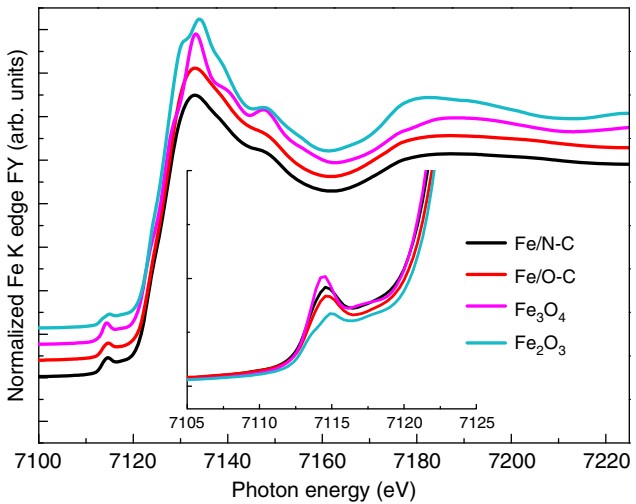

**Fig. 1** Fluorescence yield (FY) Fe K edge XANES spectra of the Fe/O-C and Fe/N-C samples. FY Fe K edge XANES spectra in comparison with $Fe_2O_3$ and $Fe_3O_4$ reference samples, measured in transmission

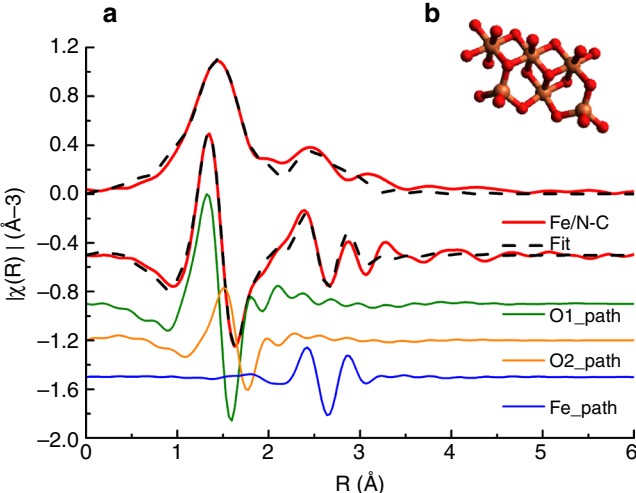

**Fig. 2** Non-phase-corrected $k^2$ weighted Fourier transform EXAFS data of Fe/N-C sample. **a** Non-phase-corrected $k^2$ weighted Fourier transform EXAFS data of Fe/N-C as an example. The imaginary part of the data, fit (using scattering paths calculated from cif file PDF 00-058-0898.cif of ferrihydrite) and scattering paths are also shown to visualize the influence of each path to the spectrum; **b** model of the *Fh*-FeOOH structure from PDF 00-058-0898.cif .Color code: Fe = orange, O = red

both tetrahedral and octahedral geometries. The intensity of the pre-edge features of both samples lies between hematite (only octahedral sites) and magnetite (which contains ~30% tetrahedral sites), implying that these samples contains some tetrahedral character, which is in agreement with the structure of ferrihydrite proposed by Michel et al.[17].

The Fourier transform of the extended X-ray absorption fine structure (FT EXAFS) data were fitted using the scattering paths calculated from a cif file of *Fh*-FeOOH (PDF 00-058-0898 from PDF-4+ structure) using two O paths and one Fe path. The magnitude of the Fourier transform of the $k^2(\chi)$ data of the EXAFS region for the Fe/N-C, reported in Fig. 2 as example, is also consistent with the structure of *Fh*-FeOOH. The two Fe–O paths are separated by a distance of 0.15 Å, and the Fe path is at 3.05 Å from the absorber atom. The Fe–O coordination number is ~6.

Fe K edge EXAFS structural parameters derived from the fits of the samples investigated reveal only negligible differences among the samples (Supplementary Table 1). Surface elemental composition of the samples is obtained by XPS at the O1s, N1s, C1s, and Fe2p core levels, by collecting electrons with kinetic energy of 450 eV corresponding to an information depth of 1.5 nm[18] (Supplementary Table 2). Notably, the O1s XP spectra in Supplementary Fig. 3 indicate the presence of $O^{2-}$ species as well as $OH^-$ species, whereas the N1s XP spectra reveal the presence of small amounts of N impurities, intuitively from the Fe nitrate precursor, present at a binding energy typical for Fe–N bonds[19]. Additionally, pyridine-like C-N species[14] are present only on the freshly prepared Fe/N-C material. The surface sensitive C1s spectrum of the Fe/N-C shows much higher intensity than the Fe/O-C at binding energy higher than the graphite-like peak (284.4 eV), indicating a higher abundance of exposed surface functional groups.

The scanning electron micrographs (SEM) reveal features characteristic to each sample: a thin layer of Fe-phase covers the surface of the fibers (Supplementary Fig. 4), whereas the amount of the bigger agglomerates and their size increase with increasing loading (Supplementary Fig. 5).

TEM images of the Fe/N-C (Supplementary Fig. 6) show that regardless of the particles size, the films or particles are

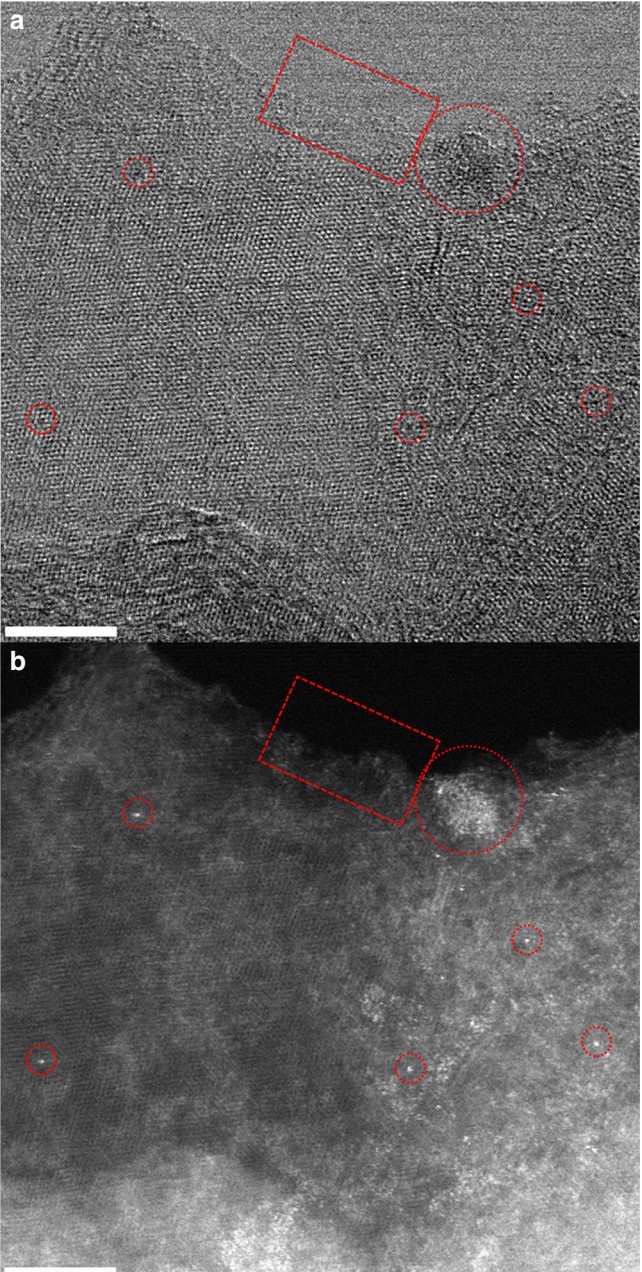

**Fig. 3** Nanostructure of Fe/N-C sample. **a** Representative Bright Field and **b** HAADF STEM micrographs of sample Fe/N-C (scale bar 5 nm). The brightest spots in the HAADF STEM image are Fe atoms. Several morphologies are identified: Nanoparticles (circle); polyatomic species (rectangle) and single atoms (small circle)

polycrystalline and composed of small agglomerated crystallites. The diffraction patterns are consistent with the *Fh*-FeOOH structure (Supplementary Fig. 6c). Note however that the off-line X-ray diffraction measurements on these catalysts failed to detect any diffraction peaks related to any Fe oxide phase, demonstrating that there is no long-range order in these materials. The similar *Fh*-FeOOH nanostructure of these samples is consistent with a condensation mechanism of small clusters[20] to form various morphologies and sizes.

Figure 3 shows a top view bright field (left) and high-angle annular dark field scanning transmission electron micrograph (HAADF-HSTEM) (right) for the Fe/N-C. Note that in the latter image, the heavier elements (in this case Fe) appear brighter.

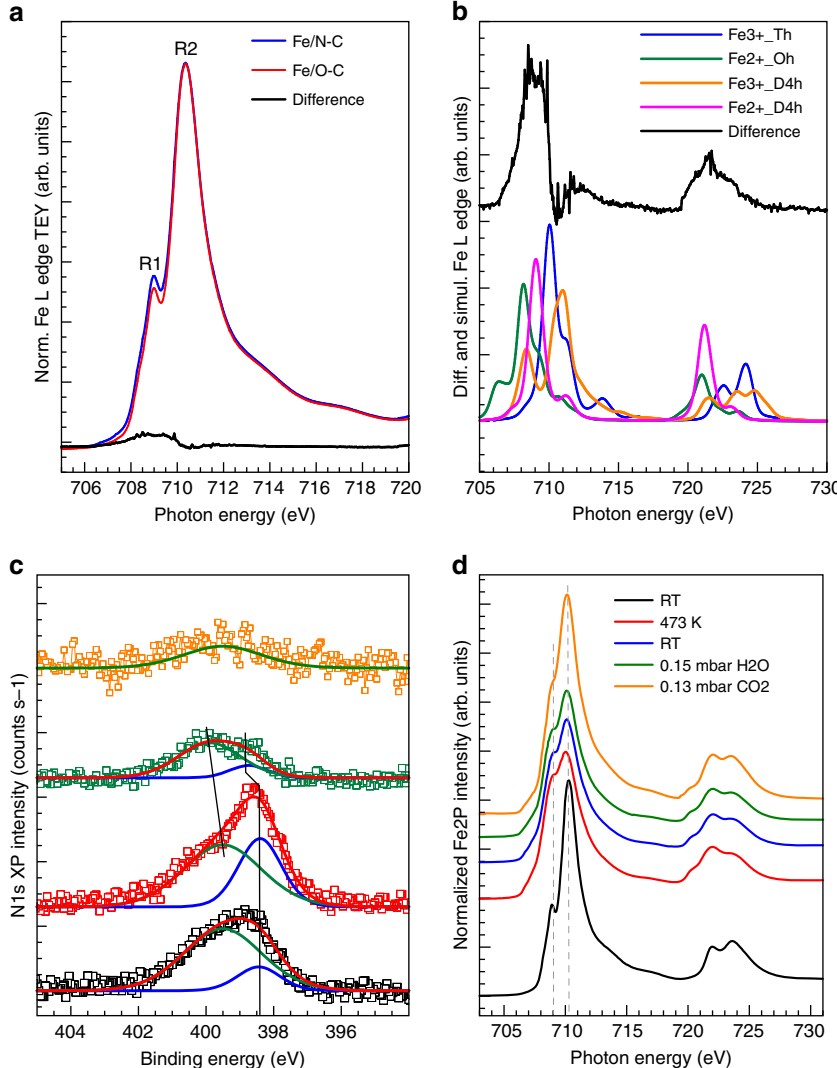

**Fig. 4** Surface sensitive XPS and NEXAFS spectra of the Fe/O-C and Fe/N-C samples. **a** Fe L edge NEXAFS spectrum of Fe/O-C (red line), Fe/N-C (blue line), and residual of the difference between the two spectra (black line); **b** Difference spectrum (black line) and simulated spectra using the CTM4XAS software[22] using simulation parameter as reported in ref. [23]. **c** Deconvolved N1s XP spectra (KE = 150 eV) using a peak for pyridine species (N-C) at 398.4 eV and full width at half maximum (FWHM) of 1.4 eV (blue line) and a peak for Fe–N species at 399.6 eV and FWHM of 2.7 eV (green line). **d** Fe L edge NEXAFS spectra relative to Fe/N-C at: 298 K in UHV (black line); 473 K in UHV (red line); after cooling at 298 K in UHV (blue line); at 298 K in 0.15 mbar $H_2O$ (green line); at 298 K in 0.13 mbar $CO_2$ (orange line)

Most interesting and only visible because of the high atomic resolution achieved in these measurements, besides bigger 3-D particles characterized by a darker contrast in the bright field HRTEM image, are the edges of the graphitic layers or the vacancies, which are decorated with Fe in clusters of atoms or as single atoms, respectively. Note that the edges of the graphitic layers are the location of the N or O species. High morphological heterogeneity, spanning from single atoms to nanoparticles, is a common feature of both samples, which makes quantitative determination of the particles size distribution impractical.

The surface sensitive Fe L near edge X-ray absorption fine structure (NEXAFS) spectra in Fig. 4a measured in ultra-high vacuum (UHV) condition reveal an important difference between these two samples. Particularly, the Fe $L_{2,3}$ edge spectra are dominated by the resonances R1 ($2p \rightarrow 3t2_g$) at 709 eV and R2 ($2p \rightarrow 3e_g$) at 710.5 eV, which are characteristic of Fe(III) species[21]; however the resonance intensities below 710 eV differ between the two samples. To assess the nature of the structural difference, spectra were simulated (Fig. 4b) using CTM4XAS

software[22,23]. Accordingly, those resonances are a signature of Fe (II) species in square planar (magenta line) or octahedral geometry (green line) on the fresh Fe/N-C sample.

A temperature-programmed XPS experiment was used to further characterize the surface chemistry of the Fe/N-C sample (Fig. 4c, d). The thermal annealing of this sample up to 473 K in UHV leads to the decrease of the O and Fe abundances and the increase of the N and C abundances (Supplementary Fig. 7). The decrease in the amount of O species is due to the condensation of the Fe oxyhydroxide structural units, while the decrease of the amount of Fe is the consequence of the particles size increase due to sintering, which are then not entirely probed by this surface sensitive measurement (ca. 0.5 nm information depth for electron of KE 150 eV). The *Fh*-FeOOH phase changes from a predominantly Fe(III) phase to a mixed Fe(III)/Fe(II) (red line in Fig. 4c, d). Likewise, the N and C abundances increase is due to the increased exposure of the support surface upon Fe sintering. However, not only the total N abundance changes upon annealing, but also the distribution of the two species, namely

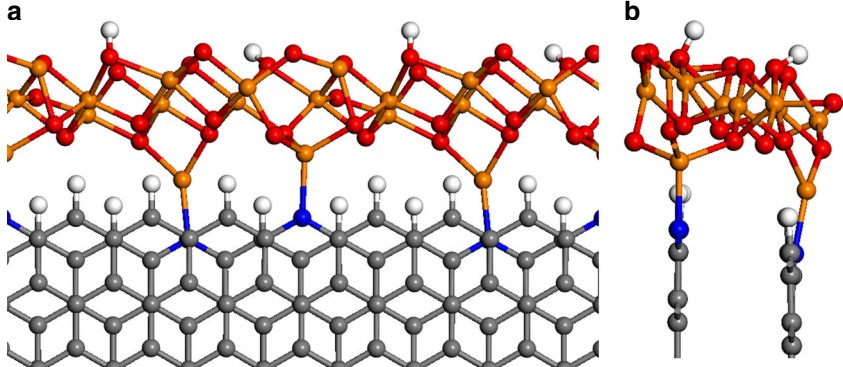

**Fig. 5** Model of the *Fh*-FeOOH/N-C interface. **a** Top and **b** lateral views of the DFT + U-relaxed geometry of ferrihydrite nanostructures decorating the N-doped graphitic zigzag edges. Color code: C = gray, H = white, N = blue, Fe = orange, O = red

N-C and Fe–N, changes. In Fig. 4c and in Supplementary Fig. 7b, c we can see that the N-C component increases significantly upon annealing. These results clearly indicate that during the impregnation step, the favorable dispersive interaction leads the Fe species in solution to adsorb preferentially on the pyridinic N species of the support and establish a chemical interaction with them as previously observed for Pd nanoparticles[24,25]. Under 0.15 mbar $H_2O$, we observe that the mixed Fe oxidation state is still stable, whereas the pyridinic N species are reduced and appears upshifted of 0.4 eV (398.8 eV), as expected for protonated species. Under 0.13 mbar $CO_2$, we observe a rapid re-oxidation of the Fe(II) species to Fe(III). Most importantly, the quantitative disappearance of the pyridinic component suggests that $CO_2$ chemisorbs not only on the Fe(II) species, but also on the pyridinic N species as previously verified by microcalorimetry[13].

In order to investigate the interaction between the graphitic edge (with and without N dopants) and the *Fh*-FeOOH-like nanostructure, we have created an idealized structural model to perform DFT simulations (Fig. 5). The iron oxyhydroxide nanostructure is assumed to have the stoichiometry and local structure of *Fh*-FeOOH. Our model is cut out from the bulk crystal structure reported by Pinney et al.[26], and is periodic in one dimension. The *Fh*-FeOOH nanostructure is assumed to be in close contact with graphite, decorating its zigzag edge, which is represented in our model by a hydrogen-terminated one-dimensional nanoribbon with two AB-stacked layers. This model is plausible considering the HAADF STEM and HRTEM images presented in Fig. 3. Along the [001] direction of the crystal, the *Fh*-FeOOH nanostructure is terminated by hydroxyl groups on one side (facing the vacuum gap) and by Fe cations on the other side (facing the graphitic edge). Along the periodic direction, we choose a supercell with eight C atoms at the edge, which minimizes the strain of the *Fh*-FeOOH layer with respect to its bulk cell parameters.

In the absence of N dopants, the interaction energy between the graphitic edge and the ferrihydrite nanostructure is calculated to be −1.1 eV for the supercell (with respect to the free edge and the unstrained ferrihydrite 1D nanostructure). However, when terminal C–H species at the edge are substituted by pyridinic N species, the interaction energy becomes significantly more negative (−1.9 eV) as the result of the formation of two N-Fe bonds per supercell. Energy minimization leads to N-Fe bond lengths of 2.0 Å and 2.1 Å, which are similar to Fe–N (pyridine) bond distances reported in the literature. It is clear then that the presence of pyridinic nitrogen at the carbon edge stabilizes the C/ferrihydrite interface, by ~0.4 eV per N-Fe bond formed. The formation of the chemical bond is accompanied by charge

transfer from the N atom to the Fe ion, whose Bader charge[27] decreases from 1.62 a.u. to 1.26 a.u. indicating a partial reduction. This is consistent with the Fe L edge NEXAFS spectrum of the Fe/N-C sample in Fig. 4a, which clearly indicates the presence of reduced Fe(II) sites as compared with the N-free Fe/O-C sample. We also tested the possibility of the formation of C-Fe bonds at the interface, via the removal of terminating hydrogen from the edge to the gas phase (in the form of $H_2$ molecules). However, this is not a favorable process as the formation of such interface, including the gas phase species, requires a large positive (5.4 eV) energy. This theoretical analysis corroborates the role of pyridinic N dopants in stabilizing the interface between graphite and the iron oxo-hydroxide particles.

**Reactivity of Fe/N-O and Fe/O-C**. The $CO_2$RR behavior of Fe/O-C and Fe/N-C samples in $CO_2$-saturated 0.05 M KHCO$_3$ solution (the pH in the bulk of the electrolyte is about 7) was investigated using the electrochemical cell depicted in Supplementary Fig. 1 and the results are compared in Table 1 at a constant cathodic potential of −0.5 V vs Ag/AgCl (3 M KCl). In this work we report the results in Ag/AgCl (3 M KCl) scale instead of the commonly used reversible hydrogen electrodes (RHE) scale.

In fact, the conversion of the applied potentials referenced to the Ag/AgCl (3 M KCl) scale into the RHE scale, taking into account the pH of the bulk electrolyte, results in $CO_2$ reduction potentials higher than the tabulated thermodynamic potentials for the $CO_2$RR to acetic acid. There are two reasons for the inapplicability of this criterion: the local pH at the surface is different than in the bulk, it may vary with time and possibly also differs widely from surface site to site[28]; and the underlying redox chemistry does not involve the $CO_2$ molecule in the gaseous state, but rather $CO_2$-related compound existing in the liquid phase such as $H_2CO_3$, $HCO_3^-$, and $CO_3^{2-}$, and, therefore, the redox potential of these species must be considered instead[29]. From the

| | **Faradaic efficiency—FE (%)** | | | | |
|---|---|---|---|---|---|
| **Catalysts** | **HCOOH** | **CH$_3$COOH** | **H$_2$** | **CO$_2$RR$^a$** | **Total$^b$** |
| Fe/N-C | 36.5 | 60.9 | 2.5 | 97.4 | 99.9 |
| Fe/O-C | 2.5 | 0 | 94.9 | 2.5 | 97.4 |

**Table 1 $CO_2$RR behavior of Fe/O-C and Fe/N-C samples at a fixed voltage of −0.5 V vs Ag/AgCl**

$^a$ $CO_2$RR Faradaic efficiency
$^b$ $CO_2$RR and HER Faradaic efficiency

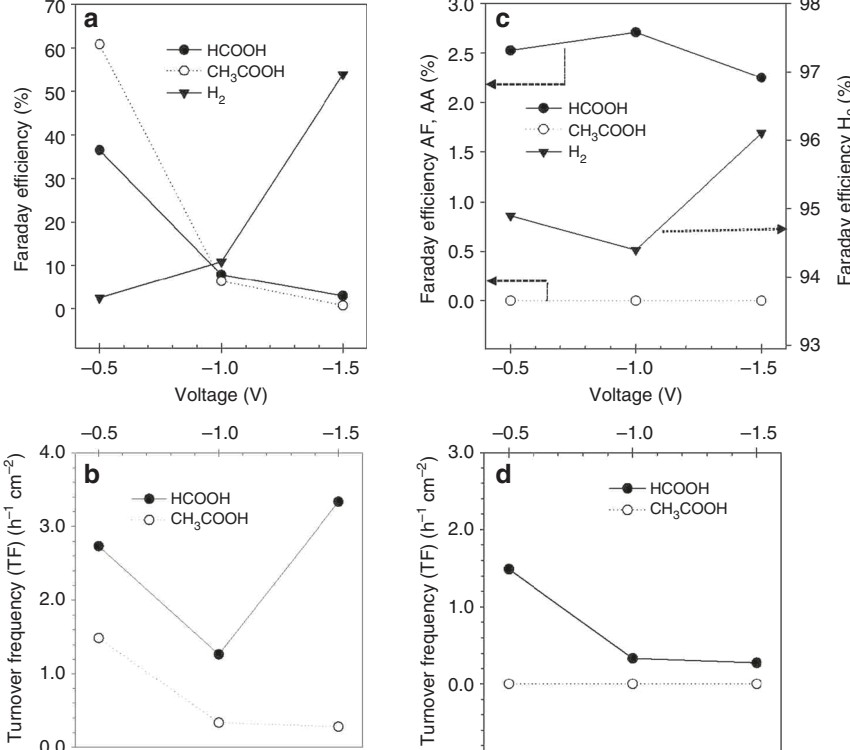

**Fig. 6** $CO_2RR$ behavior of Fe/O-N sample and Fe/O-C samples. **a** Faraday efficiency (%) to the products of $CO_2$ reduction under applied voltage of −0.5, −1, and −1.5 V vs Ag/AgCl (3M KCl) for Fe/N-C sample and **b** corresponding turnover frequency ($h^{-1} cm^{-2}$). **c** Faraday efficiency (%) to the products of $CO_2$ reduction under applied voltage of −0.5, −1, and −1.5 V vs Ag/AgCl (3M KCl) for Fe/O-C sample and **d** corresponding turnover frequency ($h^{-1} cm^{-2}$)

operando study presented later on in this paper, it will be evident that the latter point provides a better description of these experiments. Upon immersion of the samples in the liquid electrolytes, the open circuit potential drifts very quickly from circa −0.15 V to a negative value close to 0 V (−0.076 V for Fe/N-C in the example reported in Supplementary fig. 7d and −0.048 V for Fe/O-C). The anodic drift of the potential indicates that an oxidation process is taking place. This observation could be explained not only as phase transformation of some Fe(II) sites present in the solid phase[30], but also as an indication of the mobilization of the Fe(III) species of ferrihydrite in carbonate media leading to the formation of Fe(II)$_{aq}$ carbonate species[12], the latter species undergoing further oxidation and re-precipitation. However, given the small and rapid variation of the OCP, we conclude that the process is limited in this case and the ferrihydrite/KHCO$_3$ interface equilibrates rapidly.

Despite a minimal difference in the Fe nanostructures between the samples, while the HER dominates over the $CO_2RR$ for Fe/O-C under potential control, the behavior of Fe/N-C is different, with $CO_2RR$ Faraday efficiency of 97.4 %, and a much lower HER Faraday efficiency of 2.5% at −0.5 V vs Ag/AgCl. $CO_2RR$ products detected were acetic acid (60.9%) and formic acid (36.5%) (Table 1).

The Faraday efficiency for the two electrocatalysts as function of potential is depicted in Fig. 6a, c for Fe/N-C and Fe/O-C, respectively. The corresponding turnover frequency ($h^{-1} cm^{-2}$ g of product $g^{-1}$ of Fe in the electrode) for formic acid and acetic acid is shown in Fig. 6b, d.

At more negative potentials (down to −1.5 V), the $CO_2RR$ Faraday efficiency strongly decreases for the Fe/N-C, whereas the HER increases significantly (Fig. 6a). The turnover frequency passes through a minimum for formic acid, while acetic acid decreases at more negative potentials (Fig. 6b). The electrocatalytic results are fully reversible by increasing the cathodic

potential from −1.5 to −0.5 V. Average current densities of −0.36, −0.78, and −5.36 mA cm$^{-2}$ were obtained at −0.5, −1 and −1.5 V, respectively. The Fe/O-C is characterized by less pronounced changes of the catalytic performance with potential, with a very similar $CO_2RR$ Faraday efficiency, whereas the HER Faraday efficiency increases at more negative potential (Fig. 6c). The only $CO_2RR$ product with a turnover frequency that decreases at a more negative potential is formic acid (Fig. 6d). The comparative analysis of these catalysts evidences the critical role of the carbon surface chemistry and the particular benefits of the N sites, indicating that the active sites for $CO_2RR$ are located at the metal cluster/carbon interface. This is also corroborated by the poor performance of higher Fe loaded samples (Supplementary Table 3 and 4). However, at more negative cathodic potentials, Fe/N-C also becomes poorly $CO_2RR$ efficient with strong reduction of the acetic acid turnover frequency, whereas the formic acid turnover frequency reaches a minimum and then increases again (Fig. 6b).

**Structure/$CO_2RR$ performance correlation by means of operando XAFS.** Structural dynamics upon electrode polarization that correlate selectivity trends are identified by operando XAFS at the Fe K edge in fluorescence yield (FY) mode.

Accordingly, Fe/N-C and Fe/O-C behave quite differently upon polarization. The XANES spectra recorded during cyclic voltammetry (CV) (CV 10 mV/s) from open circuit potential (OCP ca. −0.1 ÷ 0 V vs Ag/AgCl in the fresh samples) to −2 V vs Ag/AgCl for Fe/N-C catalyst are shown in Fig. 7a. The corresponding current/ potential profile is reported in Fig. 7b.

At −0.5 V, the intensity of the pre-edge resonance at 7114.5 eV decreases, while the edge is now down-shifted by 2 eV. At −2 V in Fig. 7a, the Fe K edge spectrum resembles a metallic state (see

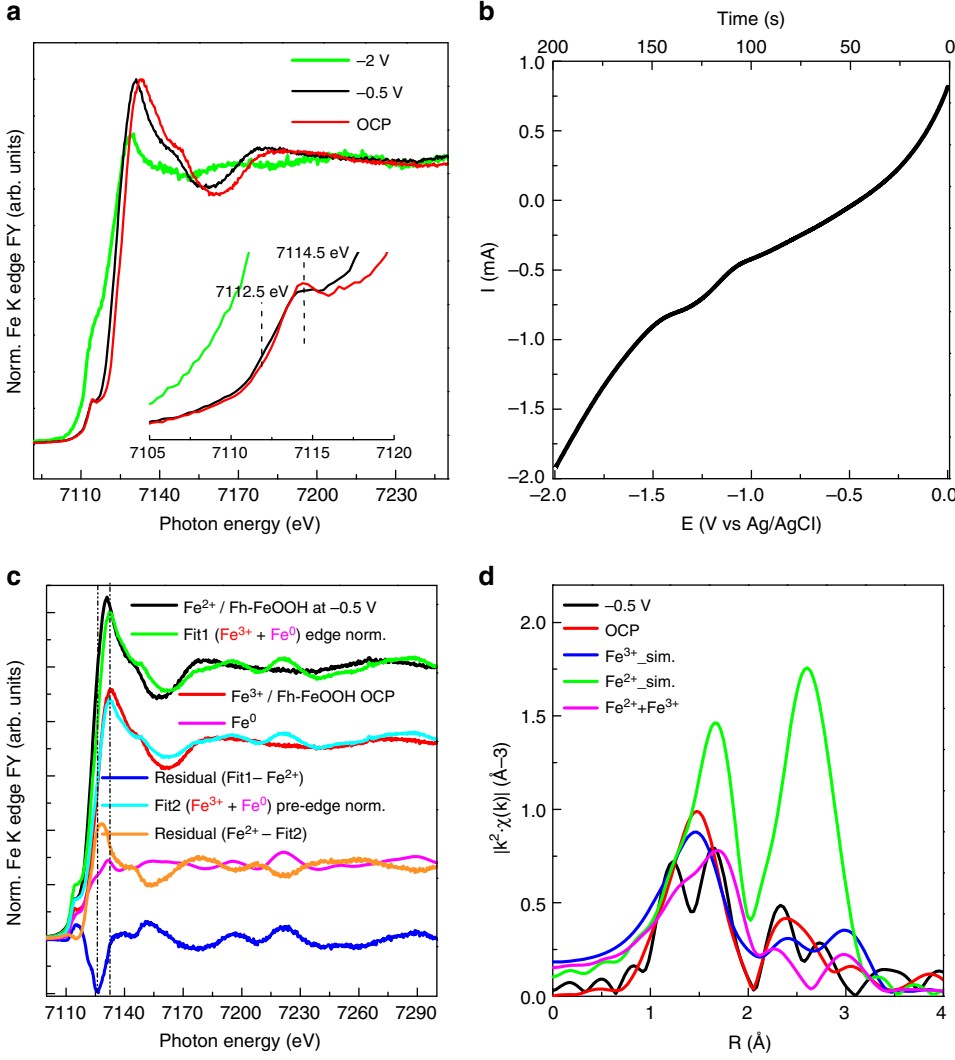

**Fig. 7** FY-mode Fe K edge XANES spectra of Fe/N-C during cyclic voltammetry. **a** FY Fe K edge XANES spectra during cyclic voltammetry and **b** corresponding voltammogram in the relevant potential region. **c** Normalized Fe K edge spectra of Fe/N-C at −0.5 V in 0.05 M KHCO$_3$ (black line); Fe/N-C at OCP in 0.05 M KHCO$_3$ component of the fit (red line); Fe foil component of the fit (magenta line); Envelope of Fe/N-C at −0.5 V with Fe$^{(III)}$ component (Fe/N-C at OCP) and Fe$^0$ component (Fe foil) in Fit 1 (green line); Difference spectrum Fit 1 - Fe/N-C at −0.5 V (blue line); Fit 1 normalized to the pre-edge in Fe/N-C at −0.5 V in 0.05 M KHCO$_3$ (cyan line); Difference spectrum Fe/N-C at −0.5 V - Fit 2 (orange line). **d** Fourier transform EXAFS spectra of: Fe/N-C at −0.5 V in 0.05 M KHCO$_3$ (black line); Fe/N-C at OCP in 0.05 M KHCO$_3$ (red line); simulations of EXAFS signal for *Fh*-FeOOH (blue line) from (PDF 00-058-0898).cif, wüstite (green line) from (amcsd_0002758) and a mixture of *Fh*-FeOOH and wüstite 1:1 (magenta line) performed with Artemis software (IFEFFIT package) using Feff6.0 code[32]

Fe foil reference spectrum in Fig. 7c). The nature of the underlying reduction process at −0.5 V resulting in the energy shift of the edge can be discerned through the linear combination of the spectrum using two components: the spectrum of Fe foil and the initial spectrum of the electrocatalyst as reference for Fe$^0$ and Fe(III), respectively (Fig. 7c) [15]. More in detail, in Fig. 7c, a fit (Fit1 green line) of the spectrum measured at −0.5 V using the spectrum at OCP in 0.05 M KHCO$_3$ as Fe(III) component (in Fig. 7a, red line) and Fe$^0$ component (Fe foil/magenta line) leaves a negative residual intensity (blue line), which clearly shows the down-shift of the edge in the region assigned to Fe(II), while the positive residual pre-edge intensity indicates that the metallic contribution is overestimated within this fit. The normalization of the Fit1 spectrum to the pre-edge intensity of the Fe/N-C at −0.5 V in 0.05 M KHCO$_3$ (in Fit 2/cyan line) leaves a residual line (orange line), which resemble the spectrum of FeO (wüstite)[31].

Changes in coordination geometry of the iron sites upon polarization were also assessed by means of FT EXAFS analysis (Fig. 7d). In order to explain the visible structural changes of the initial *Fh*-FeOOH phase upon polarization at −0.5 V, simulations of EXAFS signal were performed[32]. for: Ferrihydrite (Fe$^{3+}$_sim. blue line in Fig. 7d) as a weighted average of the contribution from three sites (Supplementary Fig. 8); Wüstite structure as signal of Fe(II) ion (green line in Fig. 7d); A mixed valence Fe(II)/Fe(III) compound formed of 50% ferrihydrite and 50% wüstite (in Fig. 7d, magenta line).

Similarly to the dry Fe/N-C /(Fig. 2), the data for the Fe/N-C at OCP in 0.05 M KHCO$_3$ (red line) fits very well the structure of *Fh*-FeOOH. We refer here to O ligands as -(O, OH) ligands being the O species present in significantly higher amount (20 at%) than N species (0.5 at%) (Supplementary Table 2).

The spectrum for Fe/N-C at −0.5 V (black line) resembles qualitatively the simulated spectrum of a mixed valence Fe(II)/Fe

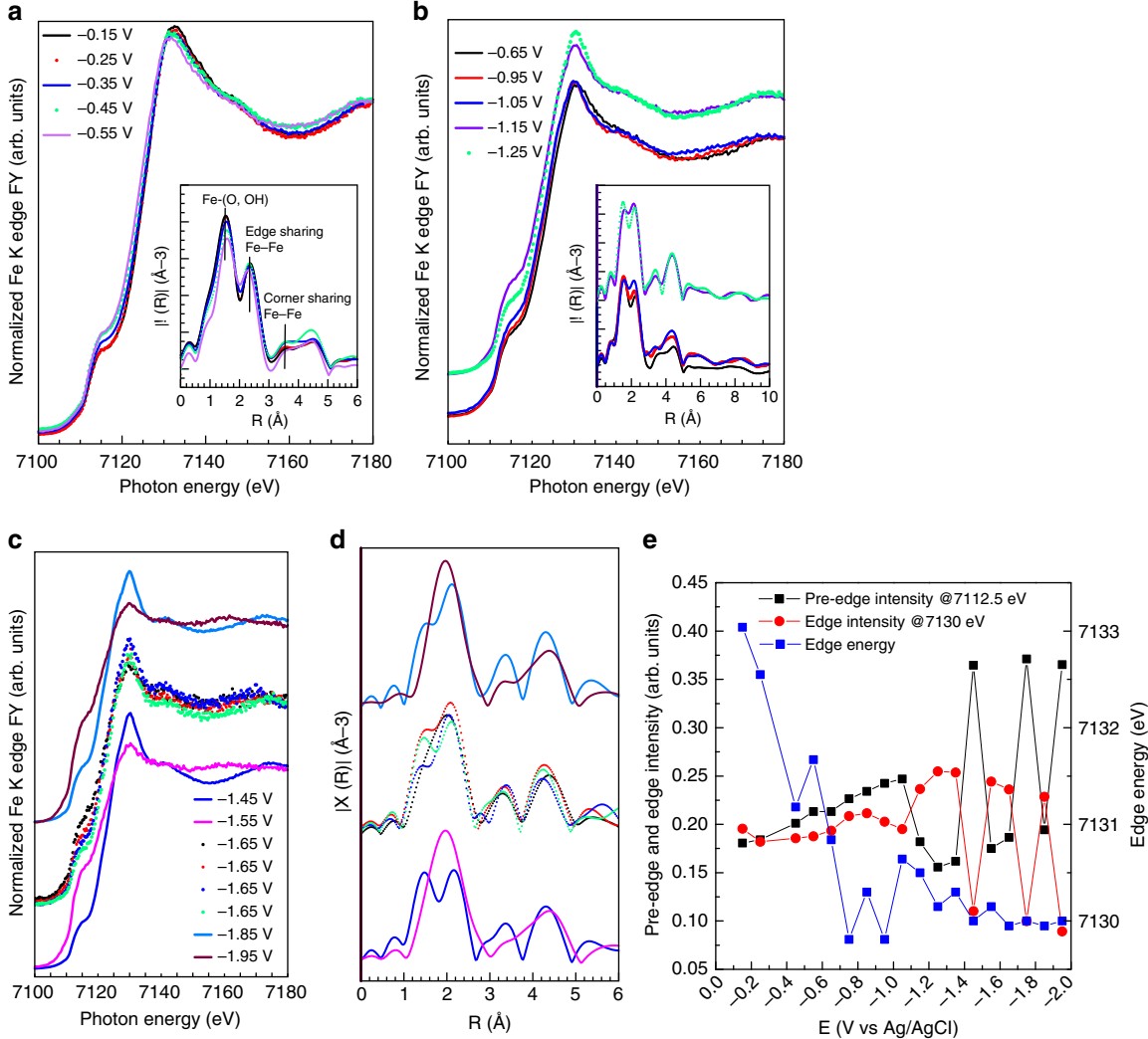

**Fig. 8** FY-mode Fe K edge XANES spectra measured in quick mode in 0.05 M KHCO$_3$ at different potential and corresponding EXAFS spectra (inset) for the Fe/N-C. **a** CO$_2$RR selective region and **b** HER selective region. **c** XANES spectra measured at high HER rate and **d** corresponding EXAFS spectra. Each spectrum is merged from 11 fast Fe K edge XANES spectra. The potential was varied in steps of 0.1 V and hold at constant potential for 3 minutes. **e** Pre-edge intensity at 7112.5 eV, edge intensity at 7130 eV, and edge energy plotted as function of the potential

(III) compound in the characteristic bimodal distribution of Fe-(O, OH) bonding lengths.

This is a consequence of the convolution of constructive and destructive interference between the signals of the different phases. We conclude that −0.5 V, a reduction of some of the Fe (III) to Fe(II) species may explain the peculiar radial distribution function. Indeed, the formation of a mixed valence compound from ferric oxyhydroxide was already reported at this potential range in 0.2 M HCO$_3^-$/CO$_3^{2-}$ solution and characterized as a Fe (II)/Fe(III) hydroxi-carbonate green rust compound[33].

After re-oxidizing the electrode at +0.77 V, quick XANES spectra were continuously recorded upon constant potential (crono-amperometry (CA)) in order to observe more detailed structural dynamics, in addition to stability. The results are reported in the Fig. 8a–d, and the current density recorded at each potential is plotted in Supplementary Fig. 9b. The key spectroscopic features such as pre-edge intensity at 7112.5 eV, edge intensity at 7130 eV and edge energy are plotted as functions of potential in Fig. 8e. In the potential region relevant to the CO$_2$RR between −0.15 V (OCP) and −0.55 V (XANES spectra in Fig. 8a), the structural dynamics observed consist mainly of an energy down-shift of the edge whereas the intensity of pre-edge

and edge does not change significantly (Fig. 8e). This behavior is consistent with the CV experiment in Fig. 7a, indicating the reversibility of the Fe(III)/Fe(II) process. In the radial distribution function, (inset in Fig. 8a) the increase of the Fe-(O, OH) distance (+0.1 Å) is also here consistent with a reduction to Fe(II) species[34]. The spectra resemble a mixed valence Fe(III)/Fe(II) phase[31], note, however, that the sample now clearly contains a small Fe$^0$ component from the previous reduction step (non-phase-corrected Fe-Fe distance 4.5 Å in inset in Fig. 8a).

In the potential window between -0.65 V and -1.25 V (XAFS spectra in Fig. 8b), HER becomes dominant. The slight increase of the pre-edge at 7112.5 eV indicates the formation of Fe$^0$. Notable changes occur at −1.15 V when the resonance at ~7130 eV becomes particularly intense whereas the pre-edge intensity decreases.

The FT EXAFS spectrum indicates that Fe–O species are still present. In the potential region from −1.45 to −1.95 V (Fig. 8c, d), the catalyst is very unstable, whereas the HER rate is at its highest. At −1.55 V, the Fe K edge XANES and FT EXAFS spectra resemble the spectra reported for Fe$^0$. The instability of the catalysts was observed several times during the CA, starting

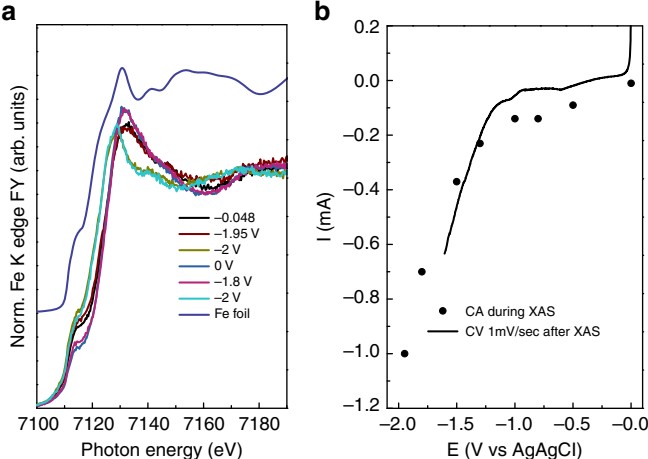

**Fig. 9** FY-mode Fe K edge XANES spectra measured in quick mode for the Fe/O-C (Fe loading 1 wt. %) at different potential. **a** Edge normalized, merged XANES spectra measured in 0.05 M KHCO$_3$ at different potential as indicated and **b** corresponding instantaneous current measured at constant potential (CA) as function of the potential itself

from −1.25 V, as a rapid switch between a Fe$^0$ state and a Fe oxidized state characterized by an unusual high intensity of the resonance at ca. 7130 eV (energy shift consistent to Fe(II)). This can be related to the HER mechanism. Despite the observation of O species in the FT EXAFS spectrum, such high intensity implies a different chemical environment than the Fe$^{(II)}$-O species observed at less negative potentials. A similar pronounced increase of the white line intensity was observed for Fe-Fe hydrogenase upon hydride bond formation and protonation[35]. Analogously, we can assume that a similar situation occurs here. At more negative potential, the higher availability of e$^−$ may lead to the formation of Fe-hydride species from OH dissociation[36]. We propose that the intermediate hydride species and adsorbed H$^+$ are discharged as H$_2$, leaving Fe$^0$ behind; however this hypothesis requires further investigation.

A different behavior was observed for the fresh Fe/O-C in operando (Fig. 9 and Supplementary Fig. 11).

Despite a similar trend and value in the OCP (the OCP drift from −0.15 to −0.048 V), the corresponding spectrum at OCP is clearly different from the spectrum relative to the fresh sample (Fig. 1), and is characterized by a pre-edge dominated by the metallic component.

The spectra measured remain otherwise unchanged as one proceeds towards more negative potentials. Structural modifications happen at −1.95 V, consistent with a further reduction of Fe (III) species to Fe$^0$ (slight increase of the pre-edge). At −2 V a 3 eV down-shift of the edge is observed and the spectrum is now similar to the one reported for wüstite[31] .The electrode is then electrochemically re-oxidized and subsequently another cycle of several CA at different constant potential (from 0 to −2 V) confirm this redox structural behavior. On the one hand, the reduction of Fe(III) to Fe$^0$ occurs for a minority of the Fe(III) species at a potential close to the thermodynamic potential ($E^0_{Fe^{3+}/Fe^0} = -0.204$ V); on the other hand, the Fe(II)/Fe$^0$ reduction of the majority of the Fe species in the sample occurs at much higher overpotential ($E^0_{Fe^{2+}/Fe^0} = -0.64$ V vs Ag/AgCl). The latter phenomenology is expected for loosely immobilized FeOOH particles or for bigger particles for which the poor electron conductivity of the ferrihydrite phase results in the observed reduction overpotential. Additionally, interfacial electron transfer resistance can be induced by the O species on the C support. On the contrary, the initial reduction of the Fe(III) to Fe$^0$

is considered being the result of the instability of some of the immobilized ferrihydrite particles in KHCO$_3$ solution. If the Fe (III) species have lost their coordination due to the thermal desorption of the oxygen species during the catalyst preparation (C1s spectrum in Supplementary Fig. 3d), they may be more susceptible to undergoing dissolution and reduction/precipitation by interacting with the reduced carbon surface. This may happen only for the very small clusters, whereas for the bigger clusters, only the interfacial Fe sites will be affected.

Operando XAFS results on a sample with higher Fe loading on O-C (20 wt. %) (Supplementary Fig. 9b and c) further corroborated this finding.

Particularly, structural changes occur significantly for this sample only above −0.8 V, with a gradual down-shift of the edge until −1.8 V when the spectrum resembles that one reported in literature for wüstite[31] .Thus, changes in the spectra are for this sample not related to either the CO$_2$RR, the HER and any structural changes accounting for the current voltage profile observed between −0.3 V and −1 V vs Ag/AgCl (Supplementary Fig. 9b), but are rather dominated by the structural dynamics of the bulk of the nanoparticles (Supplementary Figure 5).

This implies that on low loading O-C surface only the small *Fh*-FeOOH clusters are indeed reduced to Fe$^0$ at low negative potential.

## Discussion

These results clearly indicate that the carbon surface chemistry influences significantly the Fe redox chemistry with strong impact on the catalytic performance. Particularly, some of the Fe(III) sites in *Fh*-FeOOH clusters are reduced to Fe(II) on the Fe/N-C, and here stabilized against the total reduction to Fe$^0$ ($E^0_{Fe^{2+}/Fe^0}$ = -0.64 V vs Ag/AgCl) in the CO$_2$RR selective potential region. In contrast, on the O-C we observe: the total reduction of Fe(III) to Fe$^0$ for the minority of small clusters ($E^0_{Fe^{3+}/Fe^0} = -0.204$ V vs Ag/AgCl) already upon immersion in the electrolyte, which is responsible for the high FE for HER at low potentials; higher overpotentials for the Fe(III)/Fe(II) reduction of most of the *Fh*-FeOOH particles.

N dopants on carbon have a double effect: not only they coordinate CO$_2$-related species, but they also stabilize these Fe(II) species hindering their further reduction, thus inhibiting HER (at −0.5 V vs Ag/AgCl). The synthesis of acetic acid is here attributed to the existence of adjacent Fe(II) and N sites initially present on the carbon surface or formed in situ as consequence of a dissolution/precipitation of ferrihydrite in bicarbonate solution[12].

The potential range for efficient CO$_2$RR coincides with the carbonation of *Fh*-FeOOH[13] and formation of green rust, and, therefore, we postulate that the carboxylate fragment is formed as a consequence of the reduction of (bi)carbonate moieties on Fe (II) species of the metastable (bi)carbonated Fe oxyhydroxide phase or on Fe(II) single atoms, both directly interacting with the carbon surface. On a N-free carbon surface, the availability of e$^−$ and H$^+$ leads to the formation of HCOOH. On a N-C, CO$_2$-related species chemisorbed on the N atoms adjacent to Fe(II) species can undergo a 6 e$^−$ transfer to form the methyl fragment, enabling opportunities for C–C coupling between neighboring carboxylate and methyl species. As the potential is increased further, the reduction to Fe$^0$ occurs together with the HER, probably through OH reductive dissociation. We envisage that further development in hard X-ray operando valence-to-core X-ray emission spectroscopy[37,38] or soft X-ray in situ resonant valence band photoelectron spectroscopy could allow to distinguish N and O ligands at the metal center, CO$_2$-related adsorbates and how those change upon polarization and or changes in selectivity. On well-defined system such as single site catalysts,

such a study would provide a definitive clarification on the reaction mechanisms of the $CO_2$ electrochemical reduction.

In summary, the outstanding $CO_2RR$ FE (97.4%) and high selectivity to C–C coupling of Fe/N-C derive from the synergistic effect between the carbon surface chemistry and Fe–OOH nanostructure. Most relevant are the small *Fh*-FeOOH clusters or Fe single atoms at the edge of the graphitic layers, where potential induced Fe(II) species adsorb and reduce $HCO_3^-$ species. The potential at which the Fe(II) sites are formed dictates the potential for the $CO_2RR$. While the reactivity toward the formation of formic acid is related to the Fe species, N species act concertedly to enable the C–C coupling. In order to maintain these performances in an extended range of applied negative potentials, synthetic strategies must aim to maximize and further promote the stabilization of these small *Fh*-FeOOH clusters.

## Methods

**Sample preparation**. Toray™ Carbon paper TGP-H-030 (FuelCellStore.com) with thickness of 0.1 mm was cut into pieces of ~0.8 × 0.8 cm (approx. mass of 4.2 mg) and used as a support for the iron oxide particles. Prior to the metal precursor impregnation, the carbon cloth was functionalized with either O- or N-groups according to the procedure adapted from Arrigo et al.[13,39].

First, several pieces were heated to 393 K in $HNO_3$ (250 mL, 70 %, Sigma-Aldrich) for 4 h, followed by drying in static air overnight at 373 K. Oxygen functionalization with concentrated $HNO_3$ produces a hydrophilic carbon surface with mainly carboxylic functional groups. This sample is denoted as O-C. In a second step, the $HNO_3$-treated samples were put in a tube furnace under 50 mLmin$^{-1}$ $NH_3$ (99.98% Ammonia Micrographic, BOC Linde) at 873 K for 4 h. Afterward the samples were cooled down to 323 K in $NH_3$ and further to RT in $N_2$ (50 mLmin$^{-1}$, BOC Linde). A N-functionalized carbon is obtained, referred here as N-C. The Fe containing samples (Fe wt. % = 1 or 20) were obtained via incipient wetness impregnation of $Fe(NO_3)_3$•9$H_2O$ solution in $H_2O$/ethanol (24:1). An aliquot of 1 mL of a 6 gL$^{-1}$ solution was used to prepare the 20 wt % sample, whereas an aliquot of 100 mL of a 3 gL$^{-1}$ solution was used to prepare the 1 wt. % sample. The solution was added drop-wise to the single carbon cloth piece paying attention that the wetting of the carbon paper piece was homogeneous. The impregnated carbon paper pieces were dried at room temperature in air overnight. Afterward, the samples were heated at 200°C in $N_2$ (50 mLmin$^{-1}$, BOC Linde) for 3 h in order to achieve decomposition of the metal precursor without decomposition of the nitrogen species of the support. The samples were cooled down to room temperature in $N_2$ before exposure to air. The notations Fe/O-C and Fe/N-C refers to samples prepared using the O-functionalized and N-functionalized C paper support, respectively. This approach enables two main advantages: easy and direct assembling of the electrode in the in situ electrochemical cell for XAFS study and no requirement of the preparation of commonly used catalyst inks which could detach from the substrate surface into the solution with time.

**Electron microscopy techniques**. Bright field (BF) and high-angle annular dark field scanning transmission electron microscopy (HAADF STEM) images were acquired with a probe corrected ARM200F at the ePSIC facility (Diamond Light Source) with an acceleration voltage of 200 keV. Measurement conditions were a CL aperture of 30 μm, convergence semiangle of 24.3 mrad, beam current of 12 pA, and scattering angles of 0–10 and 35–110 mrad for BF and HAADF STEM, respectively. SEM analysis was performed on a Zeiss Ultra SEM operating at an acceleration voltage of 1.6 and 20 keV.

**Ambient pressure XPS and NEXAFS measurements**. Ambient pressure XPS and NEXAFS measurements in the soft X-ray regime were carried out at the ISISS end station and beamline at Helmholtz–Zentrum Berlin (HZB). The freshly prepared samples from atmospheric environment were directly exposed to vacuum (10$^{-7}$ mbar) in the XPS Chamber. XPS measurements were performed applying a suitable excitation energy corresponding to a kinetic energy (KE) of the photo-emitted electrons of 450 eV (ex situ characterization in UHV) and/or 150 eV (TP-XPS experiment) for the core levels Fe2p, C1s, O1s and N1s. The energy pass Ep was set to 20 eV.

The core levels envelopes were fitted using Casa XPS software after subtraction of a Shirley background.

The fittings of the Fe2p, O1s, and N1s were performed considering as many components with Gaussian–Lorentzian line-shape as needed to describe consistently structural changes among the samples and upon temperature-programmed XPS. The fitting of the spectra was done constraining the peak position by ± 0.05 eV. The area ratios between the Fe2p3/2 and Fe2p1/2 spin orbit split transitions was constrained to the theoretical value of 2:1 and the distance between the two-spin orbit split transition was 13.5 eV. Binding energies were referenced to the C1s core level at 284.3 eV measured after each core level

measurement at the same excitation energy. Quantification of the elemental composition was carried out assuming a homogeneous model distribution.

Auger Electron Yield NEXAFS spectra were recorded with an analyzer setting of 50 eV pass energy (Ep) and electron kinetic energy (KE) of 700 eV, 520 eV, 350 eV, and 240 eV for Fe L, O K, N K, and C K, respectively. The beamline setting was: exit slit (ES) 111 μm and fix focus constant (cff) 1.4 (cff = cosα/cosβ). The kinetic energy window was chosen such to avoid photoelectrons moving through the NEXAFS spectrum while sweeping the excitation energy, while broad Ep was necessary to obtain reasonable intensity. The exit slit value chosen enables an optimal compromise between high photon intensity and good spectral resolution. The higher order suppression operation mode of the monochromator was applied (fix focus constant cff = 1.4) to avoid contributions to the background in NEXAFS spectra that might complicate intensity normalization of the spectra on impinging photon flux. The same analyzer and beamline setting was used for measurements under environmental condition. The sample heating was assured by a IR-laser mounted on the rear part of the sample holder. Temperature control was realized using two K-type thermocouples. During the TP-XPS experiment, water was dosed through a dedicated mass flow controller to achieve a final pressure of 0.1 mbar. After evacuating the chamber to a pressure of 10$^{-7}$ mbar, $CO_2$ was dosed through a dedicated mass flow controller to achieve a final pressure of 0.1 mbar. During APXPS measurements, the gases composition was continuously monitored using a quadrupole mass spectrometer directly mounted onto the analysis chamber.

**XAFS measurements and electrochemical cell for operando study**. X-ray absorption experiments (EXAFS and XANES) were performed at the B18 Core EXAFS beamline of Diamond Light Source[40]. The measurements were carried out using the Pt-coated branch of collimating and focusing mirrors, and a Si(111) double-crystal monochromator. A couple of Pt-coated harmonic rejection mirrors were inserted before the first ion chamber and used to filter out photons with higher energy. The size of the beam at the sample position was ca. 1 mm (h) × 1 mm (v).

Samples were measured both in static air and operando conditions. The data were collected in fluorescence mode, by means of a 36-element solid state germanium detector (K$_{max}$ = 14), the ion chamber before the sample has been used for measurement of incoming photons (I0 filled with a mixture of 30 mbar of Ar and 1080 mbar of He to optimize sensitivity at 20% efficiency).

For the operando XAFS study we distinguish to different measurement modes: Operando Fe K edge EXAFS and fast-XANES. The operando EXAFS spectra at the Fe K edge (7112 eV) were obtained from 200 eV before the edge up to 900 eV after the edge (corresponding to 15.3 Å$^{-1}$ in k-space). The measuring time was 3 minutes per spectrum.

Operando fast-XANES was performed in quick mode with continuous movement of the monochromator in both directions and a constant step size equivalent to 0.3 eV. The spectra were obtained from 100 eV before the edge up to 300 eV after the edge (corresponding to 8.9 Å$^{-1}$ in k-space) and collected every 20 s. When indicated, 11 repetitions were acquired and then merged to obtain a better signal to noise ratio.

Data were normalized using the Athena[32] program with a linear pre-edge and polynomial post-edge background subtracted from the raw data. All XANES data were fitted with linear combination analysis using relevant spectra as reference. Fits were performed with Athena in the −20 to +30 eV range using relevant recorded spectra as reference, to describe variation in sample composition.

EXAFS fits were performed using ARTEMIS software[32]. For the fresh sample (reported in Fig. 2 and in supplementary table 1) the amplitudes and phases of two Fe–O and one Fe–Fe scattering paths were calculated from reported structure PDF 00-058-0898.cif of ferrihydrite. Moreover, the interatomic distances and Debye-Waller factors were optimized by fitting the experimental data.

An electrochemical cell adapted to the B18 beamline of the UK's Synchrotron Diamond Light Source was designed for the operando study. The scheme of the operando XAFS cell is reported in Supplementary Fig. 2 and details about experiments are reported in Supplementary Note 1. Before the operando electrochemical measurements, the Fe K edge spectra were measured as a dry sample and upon contact with the liquid electrolyte. Radiolysis was excluded on the basis of the time stability of the Fe K edge measured for the fresh sample in $KHCO_3$ at open circuit potential (OCP) for 30 minutes. This is also consistent with the work of N. G. Petrik et al.[41] showing that at the interface liquid electrolyte/Fe oxide, radiolysis is inhibited with respect to bulk radiolysis.

For the analysis of the FT EXAFS recorded in operando, we reasoned that a simulation approach was more appropriate rather than a fit due the possible presence of multiple Fe phases. Therefore, first, the most relevant single scattering and multiple scattering paths were calculated for each of the three crystallographic sites in the ferrihydrite structure. Upon a comparative analysis, a fit of the Fe/N-C at OCP in 0.05 M $KHCO_3$ (red line) using only the most abundant site 1 of ferrihydrite (site1 in Supplementary Fig. 8a,b) was used to extract the values for $S0^2$ amplitude (0.75), Fe–O (0.007), and Fe-Fe (0.010) Debye-Waller factors. Those values were fixed for each path in all the simulations. Finally, the weighted average of the contributions expected from the three difference sites was calculated. A similar approach was adopted for wüstite and for the mixed valence oxide simulations (50% *Fh*-FeOOH and 50% wüstite).

**Procedure for the electrochemical reduction of CO₂ in liquid phase**. A homemade electrochemical cell made of Plexiglas was employed for the electro-chemical reduction of $CO_2$ in liquid phase (Supplementary Fig. 1). The cell has a three-electrode configuration: the working electrode (about 0.64 cm²) was located at the cathode side, at a small distance from a saturated Ag/AgCl reference electrode to reduce the solution resistance. A commercial Pt rod (Amel) immersed in the anode compartment was used as the counter-electrode. The anode and cathode compartments were physically separated using a proton-conducting membrane (Nafion® 117, supplied by Ion Power). A 0.05 M KHCO₃ aqueous solution was used as the electrolyte both in cathode and anode compartments. To assure a uniform distribution of $CO_2$ in the cathode compartment, the electrolyte solution was introduced into an external reservoir and saturated with a continuous flow of pure $CO_2$ (10 mL min⁻¹). A peristaltic pump was used to continuously circulate the $CO_2$ saturated electrolyte solution through the cathode compartment and the external reservoir. A potentiostat/galvanostat (Amel mod. 2049 A) was employed to supply a constant bias between the electrodes.

The experiments were carried out at three different voltages (−0.5, −1, and −1.5 V), which were maintained for 30 min. Sampling from the external container was made to analyze the liquid products by Gas Chromatography-Mass Spectrometer (GC-MS, Thermo Trace 8000 A EVO, Triple Quadrupole MS, column Stabilwax) and Ion Chromatography (IC Metrohm 940 with conductivity and amperometry professional detector Vario). The gas products were detected by sampling the outlet gaseous stream and analyzed by Gas-chromatography (GC-TCD, Agilent 7890 A, column 5 A Plot). Before starting chronoamperometric experiments, CV measurements were conducted on the electrocatalysts in the potential interval 0/−2 V (vs Ag/AgCl) at a scan rate of 10 mVs⁻¹.

**Computational methods.** The Vienna Ab Initio Simulation Package (VASP)[42,43] was used to carry out quantum mechanical calculations within the Kohn-Sham implementation of the DFT. The Perdew-Burke-Ernzerhof (PBE)[44,45] version of the generalized gradient approximation (GGA) was employed as the exchange-correlation potential. A Hubbard-type correction was applied to Fe $3d$ orbitals following the GGA + U formulation by Dudarev et al.[46], where a single parameter $U_{eff}$ determines the strength of the correction. The GGA + U approach penalizes the $d$ orbital hybridization with the ligands, thus opposing the GGA tendency to over-delocalize orbitals. Previous work has shown that $U_{eff} = 4.0$ eV leads to optimal results in the description of the electronic structure of iron oxides[47–49]. The interaction of the valence electrons with the core was modeled using projector augmented wave (PAW) potentials, where levels up to 1 $s$ in C, N, and O and up to $3p$ in Fe were kept frozen at the atomic reference states. The number of planewaves in the basis set is controlled by the cutoff energy, which in our calculations was $E_{cut}$ = 520 eV, 30% above the standard value for the set of PAW potentials. Integrations in the reciprocal space were performed using a fine grid of Γ-centered $k$-points with a maximum separation of 0.01 Å⁻¹ in the reciprocal space. All precision para-meters were tested for convergence of the total energy to within 1 meVatom⁻¹. Spin polarization was allowed in the simulations of iron-based systems, and the magnetic moments were calculated in ferromagnetic configurations for simplicity.

**Data availability.** The authors declare that all data supporting the current findings of this study are available in the main manuscript or in the Supplementary information. Other data are available from the corresponding author on reasonable request.

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

## Acknowledgements

Dr. June Callison of the Catalysis Hub is gratefully acknowledged for help with setting up the N-C synthesis experiments and Jamie Grindrod, Stewart Scott and Phil Robbins for the design and manufacturing of the electrochemical cell for operando XAFS. We thank Diamond's Electrochemistry Project for resources and support with commissioning the in situ cell. We thank Diamond Light Source and the UK Catalysis Hub block allocation award for beamtime (SP17031) and (SP10306). The UK Catalysis Hub is kindly thanked for resources and support provided via our membership of the UK Catalysis Hub Consortium and funded by EPSRC (portfolio grants EP/K014706/1, EP/K014668/1, EP/K014854/1, EP/K014714/1, and EP/I019693/1). We thank HZB for the allocation of synchrotron radiation beamtime (15202970ST). We acknowledge STFC for financial support (grant code ST/N002385/1).

## Author contributions

The manuscript was prepared through the contribution of all authors. The contribution of UniME/INSTM/CASPE unit (C.G., G.C., S.P., and C.A.) refers to the development, design of the experimental apparatus for $CO_2RR$ process, and testing of the electrocatalysts for the evaluation of productivity and Faradaic efficiency/selectivity. The contribution of the DLS/UK Catalysis Hub unit refers to: the development of the in situ methodology for XAFS study of electrochemical processes (R. A., G. C., D. Gianolio); performing ex situ and operando XAFS measurements (R. A., G.C., D. Gianolio, D.G., V. S.), and data analysis (D. Gianolio, R.A., P.P.W., E.K.G.). R.A. conceived and supervised the study. S.K.C. prepared the electrocatalysts. M. E. S. executed the TEM and SEM experiments and evaluated the data. R. A. drafted the manuscript. G.C, C.G., P.P.W, G. H., D. Gianolio, G. Cibin, and E.K.G. gave general advisory in finalizing the manuscript. V.P. and R.G.-C. performed the DFT study. R. A. and J. V. J performed the APXPS study.

## Additional information

**Competing interests:** The authors declare no competing financial interests.

