## [Peer Review File · Nature Communications]

Reviewers' comments:

Reviewer #1 (Remarks to the Author):

"Operando XAFS study of the electrocatalytic CO₂ reduction over FeOOH on N-doped carbon" is an interesting paper on the electroreduction of CO₂ to acetic acid at quite low overpotentials on low-loading Fe oxyhydroxide deposited onto N doped carbon paper.

The catalyst is studied by XAS under operative conditions to demonstrate its stability compared to another material, where graphite is O-doped. While the N-doped one stabilizes Fe²⁺, O doped leads to metallic iron even at low potentials.

These conclusions are well demonstrated by the shown experimental results and this, together with the impressive properties of the material, make the manuscript worthy of being accepted in Nature Communications. I believe that the results and conclusions of this paper will be of interest for a large community of scientists and will inspire them for further researches.

The experimental section is well detailed.

I found the following minor issues that should be considered by the Authors:

-Fig 3 it is very hard to see the Fe "chains". The Authors should provide a better image or highlight what they want to show

-The use of too many acronyms sometimes makes reading complicated. For example: "...very similar CO₂RR FE whereas the HER FE increase at more negative potential (Fig. 4c). The only CO₂RR product is FA with a TF..."

-Referring to the sentence: "Upon immersion of the sample in the liquid electrolytes, the OCP shifts very quickly from -0.1 V circa to a negative value of -0.048 V." The Authors should provide a recording of this experiment. This is important for other scientists who might want to reproduce the results.

Reviewer #2 (Remarks to the Author):

This paper synthesized a novel FeOOH/N-C catalyst for the electrochemical reduction of CO₂, which converted CO₂ to acetic acid at a very low overpotential with excellent Faraday efficiency and selectivity. To the best of my knowledge, few researches about Fe-based catalysts in CO₂RR field have been reported, so this work pushes the field one step ahead. Another highlight of this paper is to identify the active species which are correlated with the C-C coupling through the rational design of operando XAFS and QXAFS measurements.

My first major comment is about the conclusion on page 5 row 105-107: "the sites responsible for the C-C coupling are characterized by an ensemble of Fe and N atoms which provides chemisorption sites for CO₂-related species". But, the whole paper didn't describe in detail the relationship between Fe and N atoms in terms of structure or electronic state, and the ex-situ EXAFS fitting results (Supplementary Table 1) of Fe/N-C showed that Fe atoms are not adjacent to N atoms, so it is confusing that how Fe and N atoms work together. If possible, authors should complement a DFT calculation to verify the synergistic effect of Fe and N atoms.

Second major comment is about "Figure 3. HAADF-STEM of Fe/N-C sample", which identified Fe species as nanoparticles, chains and single atoms. But, there was no strong evidence to exclude the possibility that single Fe atoms work as the active site and the authors neglected this species in the paper below.

Minor points:

1. The authors claimed "In the electro-catalytic CO₂ reduction reaction (CO₂RR), nanostructured Fe(III)OOH immobilized on N-doped carbon enables high Faraday efficiency (97.4%) and selectivity to acetic acid (61%) at very low overpotential (-0.2V vs Ag/AgCl)." on page 2 row 32-34. As far as I know, the current density is far less than 10 mA cm⁻² at this potential in this paper, so the usage of "overpotential" might induce misunderstanding.

2. Authors should additionally complement the XPS spectrum of Fe, N, C and O.

3. The minimum value of the k range for EXAFS fitting was too small.

4. Some typo errors should be revised in this paper.

In conclusion, this paper showed appreciated results in terms of the materials of catalysts and the rational design of operando XAFS measurements. But, the authors should explain in detail the interaction between Fe and N atoms through DFT calculation or other methods, and then revise this manuscript seriously for reviewing again.

Reviewer #3 (Remarks to the Author):

The paper by Arrigo report an operando XAS study on CO₂ reduction over iron catalysts. Main focus is the XAS investigation and establishment of a structure-activity correlation. The catalytic data is impressive and noteworthy, however the X-ray spectroscopic approach – although very valid and carried out in a very professional way – does not justify publication in Nature Communications.

The authors make extensive use of conventional XAS, which is the working horse of catalysis research. However, there are more modern approaches like HERFD-XANES and XES, which can be combined with conventional XAS to gain much deeper information into the structural mechanisms of such important reactions like CO₂ reductio. For example the postulated hydride formation could be potentially detected by VtC-XES.

Moreover, the XAS approach here is lacking some precision concerning the data analysis:

1) The authors discuss the existence of Fe(III), but use magnetite as reference, containing Fe(II), which makes also a comparison of the prepeak with the unknown sample suspected as Fe(III) nonsense.

2) How would a fit for other iron oxides look like? Although the scattering does indicate FeOOH, there are many other oxides with similar structural parameters, these have to be excluded by comparison of the fits.

3) The cyan line in figure 5 is interpreted as wuestit due to its resemblance to this compound. Frankly speaking, this residua spectrum looks like thousands of other iron spectra, and due to its low resolution such an assignment is overinterpreting the data.

4) The EXAFS data in figure 6 were obviously not fitted, but only interpreted on the basis of a shift (detected by eye) in the FT. This is a dangerous approach as, fast FT scripts of common programs assume a constant phase shift which might introduce artefacts in the position of the FT signals. Without a fit, the data can not be interpreted in a sound way.

5) What is probed by XAS is the average of all iron sites, and as the authors state, in some cases rather large particles are formed. This is a very serious issue, since with the presented data only an "average" structure- activity correlation is obtained. This can be completely different from the "active" structure – activity correlation.

Summing up, this paper is an interesting piece of work, but the results and the approach does not allow publication in Nat. Comm. I rather recommend J. Phys. Chem. C or PCCP.

Response to the referees' remarks

Following, a point-by-point response to the referees' remarks is reported. The modifications on the manuscript are highlighted in the text on a yellow background.

Reviewer 1:

I found the following minor issues that should be considered by the Authors:

-Fig 3 it is very hard to see the Fe "chains". The Authors should provide a better image or highlight what they want to show

Authors: We agree with the referee that describing the Fe species in the rectangle of Figure 3 as chains is probably overstated. However, to differentiate them from the Fe single atoms in the small circle, we are now discussing them as polyatomic species. This is because the bright spots (the Fe atoms) are close to each other so to imply a chemical bond between them. To make it clearer we have additionally included a paragraph in the manuscript that better explain the difference among the morphologies observed. This is as follow and it is highlighted in yellow in the main text.

Most interesting and only visible because of the high atomic resolution achieved in these measurements, besides bigger 3-D particles characterized by a darker contrast in the bright field HRTEM image, are the edges of the graphitic layers or the vacancies, which are decorated with Fe in clusters of atoms or as single atoms, respectively. In the bright field image, the poor contrast observed in the region where the clusters are localized by the HAADF-STEM image (area enclosed in the rectangle) as compared to the bigger particle, suggests that those may be 2-D structures decorating the edge terminations of two overlapping graphene layers of the graphitic support.

Basically, the comparison of the HRTEM and HAADF-STEM in the region inside the square shows two graphene layers, the upper one of slightly smaller area with termination decorated by Fe clusters. The bottom one extends a bit further but also partially decorated. We have included a TOC, which depict clearly the interaction of the Fe species with the edge termination of the graphene layers of the graphite support.

Here I would like to stress more, not the nuclearity of these species, but the fact that those are located at the edge of the graphene layers of the graphitic structure, where the heteroatoms are also located. Particularly important is the existence of pyridinic N species.

Also, we are reluctant to color the TEM image and change the contrast, for the reason that the contrast itself carries important information on the local electron density of the material. We would like to show the high quality of the data, where not only the single atoms are visible but also the hexagonal rings of the graphene layer are resolved.

-The use of too many acronyms sometimes makes reading complicated. For example: "...very similar CO₂RR FE whereas the HER FE increase at more negative potential (Fig. 4c). The only CO₂RR product is FA with a TF..."

Authors: This has been changed throughout the manuscript to make it more reader friendly. We are only using CO₂RR as acronyms.

-Referring to the sentence: "Upon immersion of the sample in the liquid electrolytes, the OCP shifts very quickly from -0.1 V circa to a negative value of -0.048 V." The Authors should provide a recording of this experiment. This is important for other scientists who might want to reproduce the results.

Authors: We have included the results of the open circuit potential monitored vs time upon the first immersion of the fresh catalysts in the electrolyte in Supplementary Figure 4d. This result was consistently observed in all electrochemical tests we have performed in both the in situ electrochemical cell for operando study at DLS, as well as the EC set up located at the University of Messina. We did perform OCP transient measured at every point of the constant potential CA operando study and observed instability only for the fresh sample.

Reviewer 2:

The authors should explain in detail the interaction between Fe and N atoms through DFT calculation or other methods, and then revise this manuscript seriously for reviewing again.

My first major comment is about the conclusion on page 5 row 105-107: "the sites responsible for the C-C coupling are characterized by an ensemble of Fe and N atoms which provides chemisorption sites for CO₂-related species". But, the whole paper didn't describe in detail the relationship between Fe and N atoms in terms of structure or electronic state, and the ex-situ EXAFS fitting results (Supplementary Table 1) of Fe/N-C showed that Fe atoms are not adjacent to N atoms, so it is confusing that how Fe and N atoms work together. If possible, authors should complement a DFT calculation to verify the synergistic effect of Fe and N atoms.

Authors: We agree with the referee that the manuscript would benefit from the clarification of the relationship between the Fe and N atoms. For this reason, we have extensively modified the manuscript by including a combined surface sensitive XPS, NEXAFS analysis and DFT calculation. The XPS and NEXAFS characterization is reported in Figure 4 at page 13 whereas the model resulting from the DFT theoretical simulations is reported in Figure 5. This combined analysis confirms the coordination of Fe sites to the pyridine N species, which is driven by an energy gain and induces also a charge transfer from the N to the Fe, with consequent reduction of the Fe(III). The reduction is confirmed experimentally by Fe L edge NEXAFS spectroscopy and simulated spectra in Figure 4b. Additionally state of the art ambient pressure XPS and NEXAFS spectroscopy were used to investigate the sites adsorbing the CO₂, proving that

both N and Fe²⁺ are able to coordinate CO₂-related species. We postulate that adjacent N and Fe²⁺ species enable opportunities for C-C coupling.

We wish to point out we have never stated in the manuscript that ex-situ EXAFS fitting results exclude the present of Fe-N bond. In reality Fe-N and Fe-O bonds have very similar length and due to the high heterogeneity of bonding configuration in the system, we are not able to discriminate between these two chemical bonds (see N1s XPS analysis). However we have now included a sentence to clarify this issue (page 21 line 41) as follows:

Although we know from the XPS analysis that the surface of the sample is also characterized by Fe-N bonds, it is not possible to discriminate between O and N ligands in such heterogeneous systems. Therefore, we will always refer to O ligands as -(O, OH) ligands with O species being present in significantly higher amount (20 at%) than N species (0.5 at%) (see Supplementary Table 2).

Second major comment is about “Figure 3. HAADF-STEM of Fe/N-C sample”, which identified Fe species as nanoparticles, chains and single atoms. But, there was no strong evidence to exclude the possibility that single Fe atoms work as the active site and the authors neglected this species in the paper below.

Authors: We agree with the referee that the single atoms play a role. Most importantly are those single atoms or polyatomic species coordinated at the edge of the graphitic plane, where the N atoms are located. In this version we took particular care to make this point clearer. For instance, in the introduction part page 5 line 11 it is stated:

We suggest that relevant species are most probably single atoms or clusters decorating the N-functionalized edges of the graphitic planes, initially present on the catalyst but also possibly formed in situ as consequence of the well-documented dissolution/precipitation mechanism of ferrihydrite in bicarbonate solution.

This point is repeated several times in the paper as more experimental and theoretical evidence are presented. The reviewer can find some of them highlighted on a yellow background.

Minor points:

1. The authors claimed “In the electro-catalytic CO₂ reduction reaction (CO₂RR), nanostructured Fe(III)OOH immobilized on N-doped carbon enables high Faraday efficiency (97.4%) and selectivity to acetic acid (61%) at very low overpotential (-0.2V vs Ag/AgCl).” on page 2 row 32-34. As far as I know, the current density is far less than 10 mA cm⁻² at this potential in this paper, so the usage of “overpotential” might induce misunderstanding.

Authors: We agree with the referee and therefore we have changed the sentence and instead of referring to the overpotential we are now reporting the absolute value of the potential.

2. Authors should additionally complement the XPS spectrum of Fe, N, C and O

Authors: The XPS spectra are reported in Supplementary Figure 3, whereas the results of the quantification is reported in Supplementary Figure 4.

3. The minimum value of the k range for EXAFS fitting was too small.

Authors: Normally, the minimum value of the k range is 2\AA^{-1} , however, to include a complete oscillation in k-space and have the same phase in the initial and final point of the Fourier transform, the k-window used was slightly lower (1.8\AA^{-1}). Note however that a Hanning window has been used so the weight given to the area from 1.8 to 2.3\AA^{-1} is less than 50% with respect to the signal falling in the middle of the window.

4. Some typo errors should be revised in this paper.

Authors: The paper was proof read by an English mother tongue.

Referee 3:

The authors make extensive use of conventional XAS, which is the working horse of catalysis research. However, there are more modern approaches like HERFD-XANES and XES, which can be combined with conventional XAS to gain much deeper information into the structural mechanisms of such important reactions like CO₂ reduction. For example the postulated hydride formation could be potentially detected by VtC-XES.

Authors: In order to address this point from a scientific prospective, we have included now data from state of the art ambient pressure XPS and NEXAFS which provide also information about the surface electronic structure and the chemical configuration of N species. We have indeed gained important insight into the reaction mechanism by applying Q-EXAFS techniques. Those were the information we were chasing.

From the technical prospective, we are glad the referee is making us aware of these techniques and we will chase opportunities for investigating this reaction with the above-mentioned methodologies in the future. However, despite their high potential, the application of these techniques to heterogeneous catalysis is not so straightforward and in the case of electrochemical reactions requires a very close collaboration between beamline scientist and academics to build up the peripheral facility for that. Several issues need to be considered as discussed in the review paper, PCCP 2014, 16, 13827.

- 1) Beam damage issue due to the high flux required for HERFD-XANES and XES.

- 2) Sample structural and morphological heterogeneity requires structural simulation and not always precise comparison of the experimental data with theoretical calculation.
- 3) The time resolution of these methodology is not comparable to the one achieved at B18 (30 sec) and we have experienced that it is much safer to measure several spectra in short time that only 1 over a larger period of time.
- 4) Few beamlines available for such study.

By means of Q-XAFS we learned about the time scale of the reaction. Without knowing the time scale of the transformation, we would be not able to come to safe conclusions. With this knowledge, we can now carry on with for instance HERFD-XANES and XES, however we believe that the we should follow a different approach with single site catalysis at the expenses of catalytic performance (we do know that single sites catalysis will not show this performance and stability).

Additionally, we feel we need to point out that despite we are supportive of advancing the technological development of a techniques, which is also evident from the scientific background of some of the coauthors who have and are investing plenty of time into the development of synchrotron based techniques, we wish to point out that the state of the art of a characterization technique should not be the solely criteria to judge whether a good scientific work must be published in high impact journal rather than others. On the contrary it should be evaluated on the basis of the new insights which are provided here.

Moreover, the XAS approach here is lacking some precision concerning the data analysis:

- 1) The authors discuss the existence of Fe(III), but use magnetite as reference, containing Fe(II), which makes also a comparison of the prepeak with the unknown sample suspected as Fe(III) nonsense.

Authors: We must disagree with this statement since in the Figure 1, which was already present in the originally submitted paper, we presented the spectrum of both hematite and magnetite, as reference for Fe(III) and Fe(II), respectively.

- 2) How would a fit for other iron oxides look like? Although the scattering does indicate FeOOH, there are many other oxides with similar structural parameters, these have to be excluded by comparison of the fits.

Authors: We believe that it is not necessary to include other references but on the contrary it would jeopardize the clarity of the paper making it less reader friendly and misleading. However, we acknowledge that the referee 's remark may originate from some missing information which we included now in the paper. To substantiate our point of view here below some comments.

Based on the preparation methodology we expected an FeOOH structure which we have first verified by XPS at the O1s core level due the presence of highly abundant OH species (this data is now reported in Supplementary Figure 3b). Moreover, the peculiar nanostructure and morphologies observed by HRTEM, are hinting at a Ferrihydrite structure, which proved to fit very well the FT EXAFS data.

However, we are pointing out that there is no long-range order in these materials. The lack of XRD peak (we are now including this information in the manuscript as a note but not presenting the data because they really show only diffraction peak of graphite) while XRD for crystalline Ferrihydrite should show 2-line or 6-line. Also, we have included in Supplementary figure 7, the simulation FT of the EXAFS function for the different sites 1, 2 and 3, and simulated the FT EXAFS spectrum as average of all these sites. So, it makes sense to discuss this as Ferrihydrite as the closest structure to the one found in our material.

This is the reason why fitting with other structures would not help but rather unnecessarily complicate the presentation of the data. However, this concept is now emphasized in the paper.

- 3) The cyan line in figure 5 is interpreted as wuestit due to its resemblance to this compound. Frankly speaking, this residua spectrum looks like thousands of other iron spectra, and due to its low resolution such an assignment is overinterpreting the data.

Authors: Based on the well documented chemistry of Ferrihydrite in bicarbonate solution, reported in geochemical papers dedicated to study of the C and Fe cycles (see references 13, 31, 35), at the potential reported for the transformation under investigation, ferrihydrite would be transformed in the so called green rust whose structure is not well known and possibly varies in terms of composition. Using an approximated description, it is a mixed $\text{Fe}^{2+}\text{Fe}^{3+}$ carbonated Ferrihydrite. The spectrum we are reporting is completely different from siderite FeCO_3 and resembles better the Wuestite. However this is a secondary point because we do not have the long-range order of the wuestite. The point we wish to make is that we use the Fe K edge of the wuestite as indicator of Fe^{2+} present but we did not imply that we have the wuestite structure. However, we acknowledge that the wording in the paper could be misleading and we have re-phrased it to make it clearer.

- 4) The EXAFS data in figure 6 were obviously no fitted, but inly interpreted on the basis of a shift (detected by eye) in the FT. This is a dangerous approach as, fast FT scripts of common programs assume a constant phase shift which might introduce artefacts in the position of the FT signals. Without a fit, the data cannot be interpreted in a sound way.

Authors: We have extensively worked on the paper to address this point. Considering that we have a mixture of phase, Fe metallic, Fe(III) and Fe(II) we reasoned that simulation was a more appropriate strategy than fitting. For this

reason, we included a Figure (7d) and the corresponding explanation of the results, as well as the technical description of the methodology in the experimental part. In Figure 7d we included the experimental EXAFS data at i) OCP; ii) -0.5V and the simulation of EXAFS spectrum of Fe(III) Ferrihydrite and simulation of Fe(II) taking the wuestite as example of Fe(II). We then did the simulation of the EXAFS spectrum of a structure that contains 1:1 ratio of the two of them and proved that the constructive and destructive interference due to the presence of two different phase lead to the peculiar EXAFS spectrum with an apparent bimodal distribution of atomic lengths as observed experimentally. The extent of the “deep” can vary if the composition varies. This corroborates furthermore the inapplicability of a fitting procedure for these mixed phases here. However, our data analysis is consistent with the presence of Fe²⁺ as discussed in our manuscript in our original submission.

- 5) What is probed by XAS is the average of all iron sites, and as the authors state, in some cases rather large particles are formed. This is very serious issue, since with the presented data only an “average” structure- activity correlation is obtained. This can be completely from the “active” structure – activity correlation.

Authors: The referee’s concern is correct but indeed this is why we have investigated not only the low loaded samples but also the high loaded sample and reported the data now in Figure 10. This figure was already present in the original manuscript.

The data clearly indicates a different behavior of the low loaded samples and high loaded samples, with the low loaded samples showing an electrochemical response mirrored in structural dynamics in the XAFS operando study. In contrast, despite the current density showing electrochemical reduction waves in the cyclic voltammogram and chronoamperometric study, not structural changes have been observed by operando XAFS for potential as negative as -0.8V. This is due to the fact that the big particles, being the ferrihydrite not conductive, do not feel the applied potential in the overall, but the electron transfer occurs only at the interface, which is very diluted with respect to the bulk of the material in the case of the high loaded sample. This is a clear indication that despite XAFS is an integral method, the dynamics observed at less negative potential, for the low loaded samples, are due necessarily to small particles strongly interacting with the carbon support, which are more abundant and dominate the signal. We did perform a position-scanning on the sample and observed similar qualitative data. The -0.5V potential region is the region interesting where selective CO₂RR occurs. At much more negative potentials, every particle will be reduced and the rate of H₂ evolution reaction is so high that a structure function correlation there cannot be done for CO₂RR but only for HER. There we do observe that the H₂ path become the favorite reaction path and mainly the catalyst is metallic. This was consistent in all the sample investigated.

On the basis of this discussion above, and also on the results of other characterization techniques here presented, we can conclude that the structure function correlation is very robust and solid.

REVIEWERS' COMMENTS:

Reviewer #1 (Remarks to the Author):

The Authors carefully replied to all Reviewers remarks. The present, new version of the manuscript satisfies all requirements to be published in Nature Communications

Reviewer #2 (Remarks to the Author):

I found that the revised manuscript has added the state-of-the-art ambient pressure XPS, NEXAFS and DFT data to discuss the interaction between N and Fe in detail. The authors also made rational explanation and modification in manuscript to eliminate the other confusions. Based on the results, I agree to publish this paper now.

Reviewer #3 (Remarks to the Author):

The authors made considerable efforts to improve the manuscript. However, the most important issues raised were not addressed. With the intention to publish the work in Nature Communications it is indeed reasonable that state-of-the-art methods are expected, especially since these methods would be capable of addressing the major issue as whether O or N are interacting with iron (as raised by other reviewers). Of course the methods are complex and not straightforward, but as mentioned in the first report, they would make the conclusions more reliable, and again, a publication in Nat Commun justifies such efforts as they would address important conclusions of the paper more precisely.

Similar arguments apply to the EXAFS analysis. The authors provide more models for comparison with the experimental spectra, but this implies that the used models are correct. A fitting procedure in terms of radial distribution functions would be a structurally unbiased approach.

Taking into account that the X-ray part remains mostly unchanged (despite of some additional sentences) I still CAN NOT recommend the paper for publication in Nat Commun.

Response to the referees' remarks

We thank the reviewer for the work of reviewing that has substantially improved our manuscript. For this reason, we opt in for a transparent peer reviewing system so that also this part is made available to the scientific community. Following, a point-by-point response to the referees' remarks is reported.

Reviewer #1 (Remarks to the Author):

The Authors carefully replied to all Reviewers remarks. The present, new version of the manuscript satisfies all requirements to be published in Nature Communications

Authors: We appreciate the reviewer, s contribution.

Reviewer #2 (Remarks to the Author):

I found that the revised manuscript has added the art ambient pressure XPS, NEXAFS and DFT data to discuss the interaction between N and Fe in detail. The authors also made rational explanation and modification in manuscript to eliminate the other confusions. Based on the results, I agree to publish this paper now.

Authors: We appreciate the reviewer, s contribution.

Reviewer #3 (Remarks to the Author):

The authors made considerable efforts to improve the manuscript. However, the most important issues raised were not addressed. With the intention to publish the work in Nature Communications it is indeed reasonable that state-of-the-art methods are expected, especially since these methods would be capable of addressing the major issue as whether O or N are interacting with iron (as raised by other reviewers). Of course the methods are complex and not straightforward, but as mentioned in the first report, they would make the conclusions more reliable, and again, a publication in Nat Commun justifies such efforts as they would address important conclusions of the paper more precisely.

Authors: In answer to the referee, the first point to be made here is that the FeOOH/N-C is an efficient catalyst and stable in a potential range as discussed in the manuscript. Those are obtained from a carbon support with N functional group only on its surface. FeOOH were immobilized on this support by wet impregnation and thermal annealing at temperature far below the decomposition temperature of the N species.

By means of "state of the art" soft X-ray ambient pressure photoelectron spectroscopy and DFT, not only we have proved that the interaction between the Fe and the N exists, but also that CO₂ chemisorb on the N site as well as on the Fe site if it is reduced to Fe(II). This was proved by the observed BE chemical shift

of the N1s and the Fe L edge. As outlook of this work, which we are currently pursuing by using APXPS, is a description of the interaction of HCO₃⁻ (what we call CO₂-related species) at the active centers. By the comparative analysis of the activity of Fe on O-C and N-C, we postulate that:

“The potential range for efficient CO₂RR coincides with the carbonation of Fe-FeOOH¹³ and formation of green rust and therefore we postulate that the carboxylate fragment is formed as a consequence of the reduction of (bi)carbonate moieties on Fe(II) species of the metastable carbonated Fe oxyhydroxide phase or Fe(II) as single atoms, both directly interacting with the carbon surface. On a N-free carbon surface, the availability of e⁻ and H⁺ leads to the formation of HCOOH. On a N-C, CO₂-related species chemisorbed on the N atoms can undergo a 6 e⁻ transfer to form the methyl fragment, enabling opportunities for C-C coupling between neighboring carboxylate and methyl species.”

It may well be that the different chemisorption geometry of the CO₂-related species will determine the reaction path to formate or to methyl as discussed in earlier literature. A part of theoretical studies, to the best of our knowledge, there is not such an experimental study which could pave the way toward the chemicals-oriented engineering of the catalysts structure.

There are few techniques that can address this question for instance V2C XES in which radiative decay from a specific orbital involved in a chemical bond can be discriminated with maximum chemical resolution (in case of Fe the 3p->1s relaxation channel). We are glad that the referee is suggesting this technique which we wish to explore in the near future. Being this a hard X-ray based technique, it will be still bulk sensitive, the intensity of the Kβ lines are still very weak, which requires high photon flux with possible damage of the sample. Time resolution is still an issue. For these particular samples, which contains a distribution of N species, among which, the most important being located on the carbon surface (note that we have maximized the signal of pyridine N species by collecting electron with KE of 150 eV which approximately escape from the 0.5 nm from the surface/vacuum interface) the signal could be too weak. However, we are able to put in place a synthetic strategy for materials suitable to be studied by this technique.

Resonance valence band photoemission spectroscopy is another technique that addresses the same question, which is more surface sensitive and gives also information about the occupied states below the fermi level. In few words, the excitation of the valence electrons (in the case of Fe 3p and 3d orbitals) with an energy corresponding to the adsorption edge of the ligand (N 1s->2p transition) could induce resonant enhancement of the signal of the projected partial density of states relative to the metal center (3d and 3p) in the valence band when hybridization between the N2p of the ligand and the Fe3d exist. In the case of adsorbed CO₂, it would be possible in principle to verify the chemical interaction between the N and Fe site and monitor how this change as function of the potential.

As the referee pointed out “the methods are complex and not straightforward” and we do agree that an effort should be made in order to address these questions, which is our ambition too.

It should be pointed out however, that the insertion of CO₂ between the metal-nitrogen bonds of metal organic framework was verified by soft X-ray absorption spectroscopy at the N K edge, which is in principle describing unoccupied states (T. McDonalds et al. Nature 519, 2015, 303).

However, the validation of the methodology is tightly related to the choice of the material to investigate. We believe that high density single site catalysts should be used to facilitate the spectroscopic analysis for both the soft and hard X-ray techniques. This requires a great deal of material synthesis and ideally, we wish to evaluate differences in activity to establish a structure function correlation. The drawback is that single site catalysis suffers very often from low activity and stability. It is difficult to attain both high activity stability and easy spectroscopic analysis.

In this paper we are discussing the high faraday efficiency of Fe/N-C as opposed to the poor performance of Fe/O-C. We apply fast XANES at a bending magnet beamline to attain high time resolution and minimize beam damage. We apply state of the art APXPS and DFT to identify the N-Fe interaction. We provide a detailed description of the dynamics of the Fe species upon polarization and change in selectivity from CO₂RR selective to HER selective catalysts. We are convinced, and the other referees agreed with us, that this is novel enough and worthy of publishing in this journal. From this, we can now proceed in the synthesis of model system, based on this finding and tailor the sample preparation to apply for instance HERDF-XANES, RIXS and V2C XES or state of the art ambient pressure resonant VB PES, which could add an additional information to the already relevant work presented here.

In order to address the point raised by the referee, we have included a sentence which summarizes this concept. This is as follows:

We envisage that further development in hard X-ray *operando* valence-to-core X-ray emission spectroscopy³⁷⁻³⁸ or soft X-ray *in situ* resonant valence band photoelectron spectroscopy could enable to distinguish N and O ligands at the metal centre, CO₂-related adsorbates and how those change upon polarization and or changes in selectivity. On well-defined system such as high density single site catalysts, such a study would provide a definitive clarification on the reaction mechanisms of the CO₂ electrochemical reduction

We feel, we need to point out again, that despite we are supportive of advancing the technological development of a techniques, which is also evident from the scientific background of some of the coauthors who have and are investing plenty of time into the development of synchrotron based techniques, we wish to point out that the state of the art of a characterization technique should not be the solely criteria to judge whether a good scientific work must be published in high impact journal rather than others. On the contrary it should be evaluated on the basis of the new insights which are indeed provided here.

Referee: Similar arguments apply to the EXAFS analysis. The authors provide more models for comparison with the experimental spectra, but this implies that the used models are correct. A fitting procedure in terms of radial distributions functions would be a structurally unbiased approach.

Taking into account that the X-ray part remains mostly uncanged (despite of some additional sentences) I still CAN NOT recommend the paper for publication in Nat Commun

On the contrary, we have provided further qualitative data analysis, and valid argument and demonstration in Figure 7 of the main text and Supplementary Figure 8 of the interference of the several phases which leads to deep in the radial distribution function. The fit would serve to extract quantitative data on bond length and coordination number however this was here not possible due to the too many phases present (not only the three considered qualitatively of Fe³⁺, Fe²⁺, Fe⁰) and due the too high heterogeneity of the chemical bonds in these samples, several Fe-OH, Fe-O, Fe-N, considering only the three phases discussed would not be representative of the real situation. Note that we did perform a fitting when possible (Figure 2). We do not think that such analysis would provide more insights in the reaction mechanism than what we have already provided.